# Change on the Circulation of Respiratory Viruses and Pediatric Healthcare Utilization during the COVID-19 Pandemic in Asturias, Northern Spain

**DOI:** 10.3390/children9101464

**Published:** 2022-09-24

**Authors:** Elisa García-García, Mercedes Rodríguez-Pérez, Santiago Melón García, Reyes Fernández Montes, Cristina Suárez Castañón, Mª Cristina Amigo Bello, Cristina Rodríguez Dehli, Carlos Pérez-Méndez, Mª Agustina Alonso Álvarez, Laura Calle-Miguel

**Affiliations:** 1Pediatrics Department, Centro de Salud Laviada, 33207 Gijon, Asturias, Spain; 2Microbiology Department, Hospital Universitario Central de Asturias, 33011 Oviedo, Asturias, Spain; 3Pediatrics Department, Hospital Valle del Nalón, 33920 Langreo, Asturias, Spain; 4Pediatrics Department, Hospital del Oriente de Asturias Francisco Grande Covián, 33540 Arriondas, Asturias, Spain; 5Pediatrics Department, Hospital Álvarez Buylla, 33611 Mieres, Asturias, Spain; 6Pediatrics Department, Hospital Universitario San Agustín, 33402 Aviles, Asturias, Spain; 7Pediatrics Department, Hospital Universitario de Cabueñes, 33394 Gijon, Asturias, Spain; 8Pediatrics Department, Hospital Universitario Central de Asturias, 33011 Oviedo, Asturias, Spain; 9Instituto de Investigación Sanitaria del Principado de Asturias (ISPA), 33011 Oviedo, Asturias, Spain

**Keywords:** COVID-19, respiratory virus, circulation, children, SARS-CoV-2

## Abstract

(1) Background: The COVID-19 pandemic and the implementation of restrictions and nonpharmaceutical interventions (NPIs) changed the trends in respiratory viral circulation and the pattern in pediatric healthcare utilization; (2) Methods: A retrospective, multicenter observational study designed to analyze the impact of the pandemic on pediatric healthcare utilization and the viral circulation pattern in children in a region in Northern Spain was carried out. Viral diagnostics data from all nasal or pharyngeal swabs collected in children in Asturias during the periods of March 2018–September 2019 and March 2020–September 2021 were analyzed, as well as the number of pediatric hospitalizations and emergency visits; (3) Results: A total of 14,640 samples were collected during the pandemic period. Of these, at least one respiratory virus was detected in 2940 (20.1%) while 5568/10,298 samples were positive in the pre-pandemic period (54.1%); *p* < 0.001. The detection of both enveloped and non-enveloped viruses decreased among periods (*p* < 0.001). After week 14, 2020, enveloped viruses were no longer detected until one year later, while non-enveloped viruses continued to be detected in children. Overall, a mean of 4946.8 (95% CI 4519.1–5374.4) pediatric emergency visits per month during the period 2018–2019 as compared to 2496.5 (95% CI 2086.4–2906.5) for 2020–2021 occurred (*p* < 0.001). The mean of pediatric hospitalizations also significantly decreased between periods, as follows: 346.6 (95% CI 313–380.2) in 2018–2019 vs. 161.1 (95% CI 138.4–183.8); *p* < 0.001; (4) Conclusions: Our study showed a remarkably reduction in pediatric hospitalizations and emergency visits and a change in the pattern of viral circulation during the COVID-19 pandemic in Asturias. The usual seasonal respiratory viruses, namely influenza or RSV were nearly absent in the pediatric population during the pandemic.

## 1. Introduction

The coronavirus disease 2019 (COVID-19) was declared a global pandemic on 11 March 2020, due to the rapid increase in cases and its spread throughout the world [1]. On 31 January 2020, Spain reported its first case of COVID-19 [2]. During the first wave of the SARS-CoV-2 pandemic restrictive measures were implemented worldwide, from spring 2020 onwards, with the aim of minimizing transmission.

On 14 March 2020, a total lockdown was declared in Spain [3]. Most of the nonessential activities were forbidden and domestic and international travel restrictions, as well as school closures were implemented. This state of alarm lasted until 21 June [4] and was followed by a phased easing of restrictions, to be discharged as the epidemiological situation developed [5]. Schools and kindergartens were closed in Spain from 13 March to September 2020, opening after the summer holiday. Some preventive measures were implemented, such as masks for school staff and children older than six, restriction of large group activities, and contact tracing in schools [6]. In Asturias, a region in Northern Spain, such nonpharmaceutical interventions (NPIs) as face mask use, hand hygiene, social distancing, and limits on mass gatherings were enforced at all restriction levels.

These preventive measures useful in containing COVID-19 transmission were not specific to SARS-CoV-2 and seemed to affect the circulation of other respiratory viruses. Many countries have reported changes in viral seasonality during the pandemic, with the disappearance, among others, of enveloped viruses, such as influenza or respiratory syncytial virus (RSV) [7,8,9,10,11,12,13,14,15,16,17]. Non-enveloped viruses, including rhinovirus or adenovirus, kept circulating during the pandemic. This difference in detection between enveloped and non-enveloped viruses might be related to their stability [16]. Whether these changes were caused by NPIs, viral interference, or both, is still under debate [10,18]. The decrease in respiratory viral circulation in children, in addition to public awareness of the pandemic and fear from parents to risk exposure to SARS-CoV-2 in a healthcare setting, lead to a vast reduction in pediatric healthcare utilization [19,20,21,22,23].

The aims of this study were as follows: (1) to describe the respiratory viral circulation pattern in children in Asturias during the first 18 months of the COVID-19 pandemic, (2) to compare the respiratory viral circulation between the same time period in 2018–2019 and 2020–2021, and (3) to compare the number of pediatric hospitalizations and emergency visits per month among both periods.

## 2. Materials and Methods

### 2.1. Study Setting and Population

A retrospective, multicentre observational study designed to analyze the impact of the COVID-19 pandemic on the viral circulation pattern in children and the pediatric healthcare utilization in Asturias during the first 18 months of the pandemic was carried out. Asturias is a region in Northern Spain with a total population of 1,002,097 people and 95,698 children protected by the National Health System in 2021. Asturias is divided into eight health areas with a network of primary care centers as well as a reference hospital for each of the areas. In the region, 76% of the total population is concentrated into three major urban areas, namely Avilés (III), Oviedo (IV), and Gijón (V) [24].

Viral diagnostics data from all nasal or pharyngeal swabs collected in children (up to 14 years old) in Asturias during the periods March 2018–September 2019 and March 2020–September 2021 were analyzed, as well as the number of pediatric hospitalizations and emergency visits per month and per hospital during both periods.

### 2.2. Viral Diagnostics

Viral diagnostics, except SARS-CoV-2, are centralized in Asturias and all the respiratory samples from the eight health areas were sent to the virology laboratory of the Hospital Universitario Central de Asturias, located in Oviedo. Viral diagnostics were performed for most children with acute respiratory infection (ARI). The swabs were placed in a viral transport medium (UTM viral transport, Copan Diagnostics Inc., Murrieta, CA, USA) and nucleic acids were extracted by using the automated nucleic acid purifier MagNAPure 96 System (Roche Diagnostics S.L., Rotkreuz, Switzerland) following the manufacturer’s instructions. The identification of respiratory viruses was accomplished using an in-house multiplex real-time RT-PCR that has been validated for use. Viral genomes were amplified using TaqMan^®^ Fast Virus 1-Step Master Mix (Life Technologies, Carlsbad, CA, USA). Amplifications and data analysis were performed using either a 7500 or a QS5 Real Time PCR System (Applied Biosystems, Foster City, CA, USA). The following viral pathogens were identified: influenza virus, RSV, adenovirus, enterovirus, parainfluenza virus, metapneumovirus, rhinovirus, parechovirus, and human coronavirus (NL63, OC43, 229E, HKU21). Repeated swabs from the same child collected within the same month were excluded.

In order to avoid biased data from a unique area, SARS-CoV-2 data were analyzed from the centralized regional database in Asturias [25]. Data from both 7-day-incidence SARS-CoV-2 in children under 14 years and 7-day-incidence SARS-CoV-2 in all ages per 100,000 inhabitants from March 2020 to September 2021 were collected and are represented in Figure 1. As represented, the most remarkable waves occurred in October 2020, February 2021, and July 2021.

### 2.3. Ethical Considerations

This study was conducted in accordance with the 1964 Declaration of Helsinki and its subsequent amendments. It was approved by the Asturian Research Ethics Committee (Project 2020-315) in July 2020. No personal records were handled during the study.

### 2.4. Statistical Analysis

Respiratory viral circulation was analyzed weekly for each of the mentioned viruses and compared with the evolution of the COVID-19 pandemic in the pediatric population. Moreover, the detection rate was calculated by dividing the number of positive samples among the total number of pediatric samples in a period (%), globally and for each virus. Pediatric hospitalizations and emergency visits were calculated monthly in each hospital. Descriptive analysis was performed using frequencies and proportions for categorical variables and mean and 95% confidence interval (CI) for continuous variables.

Comparisons in viral detections rates, pediatric hospitalizations, and emergency visits between two periods (March 2018–September 2019 and March 2020–September 2021) were calculated. A chi-square test for categorical variables and Student’s t-test for continuous variables were carried out for the comparative analysis. A *p*-value cut-off of 0.05 was considered statistically significant. The analysis was carried out using R software Version 4.1 (R Foundation, Vienna, Austria) and the graphs were created with Microsoft Excel Version 16.62.

## 3. Results

### 3.1. Respiratory Viral Circulation during the COVID-19 Pandemic

A total of 14,640 samples were collected during the period March 2020–September 2021. Of these, at least one respiratory virus was detected in 2940 samples (20.1%). Figure 2 represents the detection of the most remarkable respiratory viruses during the time of the study, as follows: adenovirus (*n* = 1266/8.6%), enterovirus (*n* = 370/2.5%), RSV (350/2.4%), rhinovirus (73/0.5%), and influenza (39/0.3%). As shown, the number of viral detections was lowest during the hard lockdown (March to June 2020) and it increased after, as the strict infection control and distancing measures were reduced. The COVID-19 pandemic affected the detection of most respiratory viruses. After week 14, 2020, enveloped viruses (influenza, RSV, human coronavirus, metapneumovirus, parainfluenza) were no longer detected until one year later, while non-enveloped viruses (rhinovirus, adenovirus, enterovirus, and parechovirus) continued to be detected in children (Figure 3).

### 3.2. Comparison of Respiratory Virus Detection between 2018–2019 and 2020–2021

A total of 10,298 samples were collected during the period March 2018–September 2019. Of these, at least one respiratory virus was detected in 5568 samples (54.1%). As shown in Table 1, the total detection rate by year and of every virus was significantly higher in the period 2018–2019 when compared to 2020–2021, except for parechovirus. This was especially important for influenza, which remained almost undetected during the period 2020–2021 with a detection rate of 0.3%. Adenovirus was the most frequently detected virus in both periods (21.3% and 8.6%).

Figure 4 shows the weekly detection of the most frequent isolated viruses in both groups, enveloped and non-enveloped, during the pre-pandemic and pandemic periods. As represented, most viruses showed seasonal trends during the period 2018–2019. Among the enveloped viruses, RSV was detected in fall and winter with the highest detection number the last week of 2018, just before the influenza epidemic (Figure 4a). Influenza was mostly detected in winter. During the COVID-19 pandemic, these seasonal trends changed, and it was especially remarkable for influenza and RSV. Influenza virus was no longer detected after week 12, 2020, and it remained undetected during the 2021 winter season (Figure 4b). Furthermore, RSV also disappeared at the beginning of the pandemic (week 12, 2020), reappearing at the end of the 2021 spring and reaching the highest detection number during the summer (week 27, 2021). Of the non-enveloped viruses, enterovirus disappeared only during the hard lockdown (week 12–26, 2020), circulating afterwards in lower incidence compared to 2018–2019, and peaking during the 2021 summer (Figure 4c). Adenovirus was the only virus detected all year long in both periods, but during the COVID-19 pandemic the number of detections was significantly lower when compared to the pre-pandemic period (Figure 4d).

### 3.3. Comparison of Pediatric Hospitalizations and Emergency Visits between 2018–2019 and 2020–2021

A total of 7849 children were admitted to the hospital during the period March 2018–September 2019. As shown in Figure 5a, the pediatric hospitalizations per month during the same period in 2020–2021 were much lower, with a total number of 4018. The same occurred with the pediatric emergency visits (Figure 5b) with a total number of 111,115 for the period 2018–2019 vs. 63,867 for the period 2020–2021. These differences between periods were also described in all the healthcare areas of Asturias, with the only exception being Area I, where the pediatric hospitalizations were slightly higher in the period 2020–2021 (318 vs. 271), as can be seen Figure 5c,d.

Overall, a mean of 4946.8 (95% CI 4519.1–5374.4) pediatric emergency visits per month during the period 2018–2019 as compared to 2496.5 (95% CI 2086.4–2906.5) for 2020–2021 occurred (*p* < 0.001). The mean of pediatric hospitalizations also significantly decreased between periods, as follows: 346.6 (95% CI 313–380.2) in 2018–2019 vs. 161.1 (95% CI 138.4–183.8); *p* < 0.001, as shown in Figure 6.

The pattern in pediatric healthcare utilization between periods also changed. As represented in Figure 5a,b, the lowest number of hospitalizations and emergency visits occurred in April 2020, right at the beginning of the pandemic and during the hard lockdown. During the summer season of 2020 and 2021, the number of hospitalizations and emergency visits increased, contrary to the period 2018–2019, where July and August were the months with the lowest number of pediatric admissions and emergency visits.

## 4. Discussion

This study shows how the COVID-19 pandemic changed the trends in respiratory viral circulation, as well as the pediatric healthcare utilization in a pediatric population in a northern Spanish region during the first 18 months of the pandemic. Viral diagnostics are performed in our laboratory for most children attending the hospital with acute respiratory infection (ARI). Therefore, this study provides information about a large number of collected swabs during two periods (pre-pandemic and pandemic) and allows us to have an up-to-date knowledge of the respiratory viral circulation. The number of pediatric hospitalizations and emergency visits are also described and compared between periods. This enables us to obtain an accurate snapshot of the implications of COVID-19 in our region.

In this study, we found that 20.1% of the collected swabs during the period March 2020–September 2021 were positive for at least one respiratory virus. This percentage was significantly lower when compared to the one found in the same time period in 2018–2019 (54.1%); *p* < 0.001. This is in the range of the results from other studies, which have reported from 15% to 68% of viral detections within their samples [8,9,26]. The difference between periods can easily be explained by the fact that, during the pandemic, oropharyngeal or nasopharyngeal samples were collected from most children attending the hospital, while during the pre-pandemic period only children with acute respiratory infection were tested.

During the hard lockdown, from 14 March to 21 June 2020, the viral circulation rapidly dropped and was the lowest of the pandemic period. Adenovirus was the only virus detected at this time and circulating during the whole period. Other studies have also described adenovirus as the only virus circulating during 2020 with similar characteristics to previous years, but with a lower incidence [9]. After week 27, 2020, other non-enveloped virus, such as rhinovirus or enterovirus, started circulating, while enveloped virus (such as influenza and RSV) remained undetected. This difference in detection between enveloped and non-enveloped viruses may be related to their stability and resistance to ethanol-containing disinfectant. As published, rhinovirus, as a non-enveloped virus, is relatively resistant to disinfectant and can survive on surfaces for a prolonged period of time [16,27,28].

As described in the literature, influenza was detected at the beginning of March 2020 and, after week 12, disappeared, remaining undetected during the 2021 winter season [7,8,9,12,13,14]. The same occurred to RSV, even though lockdown measures were progressively relaxed, and schools reopened in September 2020, when bronchiolitis and RSV cases were expected to re-appear. In Spain, a 7 month delayed transmission wave of RSV was observed, peaking in the first week of July [11]. This is entirely in agreement with the situation in Asturias, as RSV reappeared in May 2021 with a peak of 52 detections the first week of July.

Although it is remarkably important to study the viral circulation in children because they have usually play a significant role in the transmission to the adult population, the COVID-19 pandemic has changed the paradigm. As shown in Figure 1, the 7-day incidence peaks in the pediatric population were preceded by the adult population. Nevertheless, the change on the pediatric viral circulation pattern may predict the trends in adult population and help taking public health decisions.

Our data show a significant decrease in pediatric emergency visits and hospitalizations during the COVID-19 pandemic in Asturias. This is consistent with other reports on the reduction in healthcare utilization during the pandemic [19,20,21]. Although we can only speculate about the possible causes for this decrease, it seems likely that the combination of COVID-19 fear at the beginning of the pandemic, combined with the reduction in respiratory viral circulation, may have produced a big impact on pediatric healthcare utilization.

This study has some limitations that should be acknowledged. First, this is an observational retrospective study. Although it allows us to describe what occurred with the viral circulation and pediatric healthcare utilization during the COVID-19 pandemic in Asturias, we cannot make conclusions on the contribution of NPIs or viral interference. Additionally, clinical or other laboratory information was not collected, so we cannot study individual clinical follow-up or analyze the causes of pediatric healthcare utilization. Nevertheless, a strength of this study is the great number of samples analyzed for viral detection from June 2020 to September 2021, as well as the whole number of pediatric emergency visits and hospitalizations in Asturias. Furthermore, 2020–2021 was compared with 2018–2019, to not only describe but also to analyze the changes over the past years.

In conclusion, our study showed a remarkably reduction in pediatric hospitalizations and emergency visits, as well as a change in the pattern of viral circulation during the COVID-19 pandemic in Asturias. The usual seasonal respiratory viruses, such as influenza or RSV, were nearly absent in the pediatric population in our region during the time of the study. The reappearance of RSV transmission did not occur until the summer of 2021, outside of the typical seasonal cycle. As shown, respiratory viral circulation should be continuously monitored to guide further public health decisions.

## Figures and Tables

**Figure 1 children-09-01464-f001:**
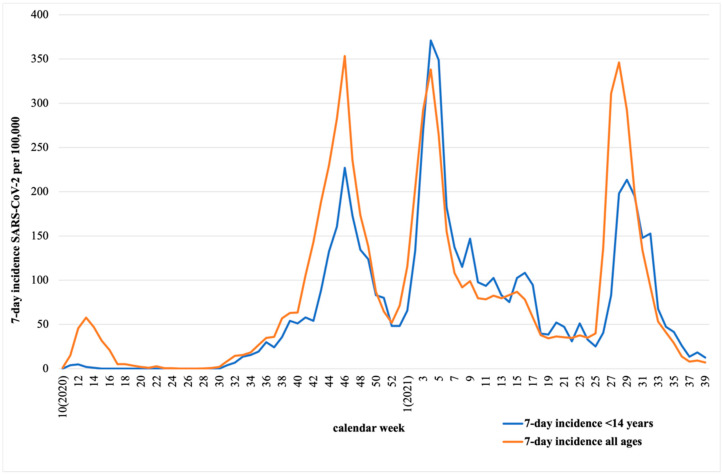
Comparison of 7-day-incidence SARS-CoV-2 per 100,000 inhabitants of the total population vs. 7-day-incidence per 100,000 inhabitants under 14 years of age in Asturias from March 2020 to September 2021.

**Figure 2 children-09-01464-f002:**
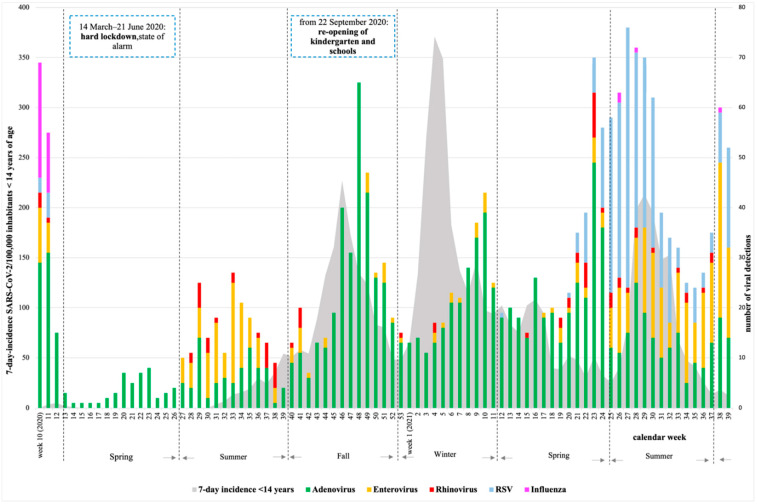
Detection of respiratory viruses among 14,640 collected swabs from week 10, 2020 to week 39, 2021 in Asturias. Depicted in gray is the 7-day-incidence SARS-CoV-2 in children under 14 years per 100,000 inhabitants in Asturias (https://obsaludasturias.com/obsa/niveles-covid-19/) (accessed on 8 June 2022).

**Figure 3 children-09-01464-f003:**
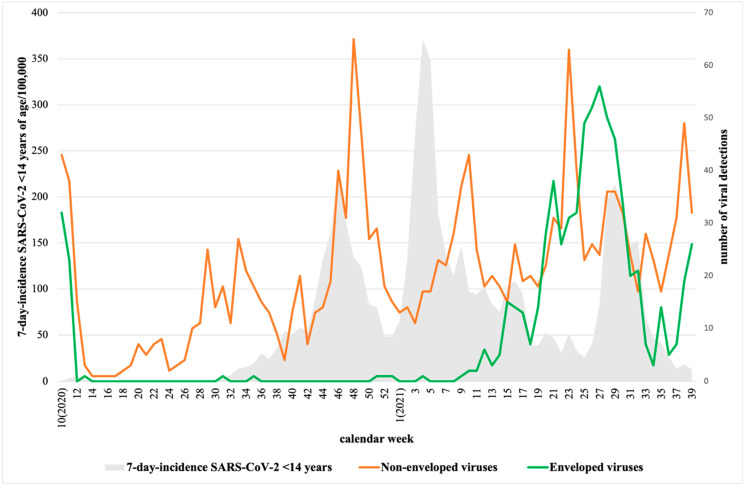
Detection of non-enveloped viruses (rhinovirus, adenovirus, enterovirus, and parechovirus) and enveloped viruses (influenza, RSV, human coronavirus, metapneumovirus, and parainfluenza) in Asturias from week 10, 2020 to week 39, 2021. Depicted in gray is the 7-day incidence SARS-CoV-2 in children under 14 years of age per 100,000 inhabitants in Asturias (https://obsaludasturias.com/obsa/niveles-covid-19/) (accessed on 8 June 2022).

**Figure 4 children-09-01464-f004:**
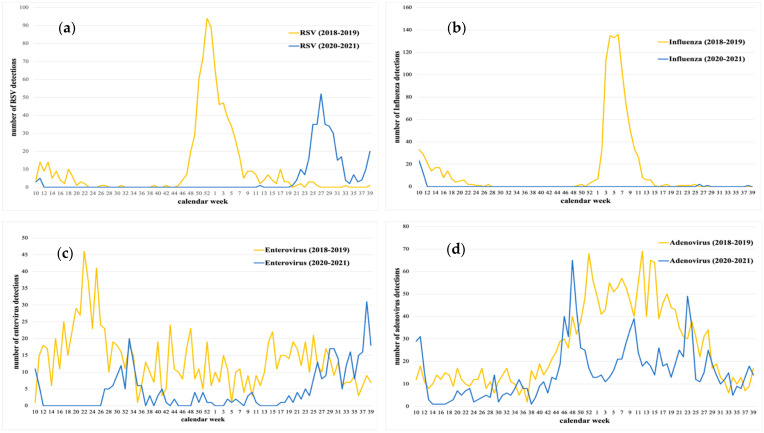
Detection comparison of weekly detection of (**a**) RSV, (**b**) influenza, (**c**) enterovirus, and (**d**) adenovirus between the periods 2018–2019 and 2020–2021.

**Figure 5 children-09-01464-f005:**
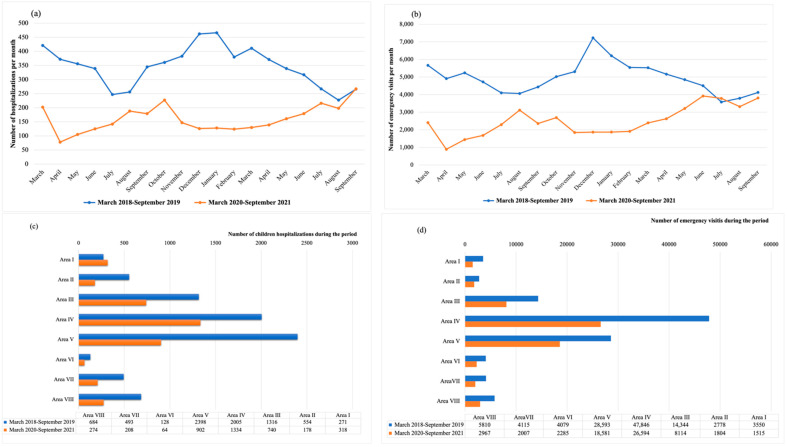
Comparison of pediatric healthcare utilization in Asturias between the periods 2018–2019 and 2020–2021. (**a**) Comparison of pediatric hospitalizations per month between periods; (**b**) comparison of pediatric emergency visits per month between periods; (**c**) comparison of pediatric hospitalizations per health area between periods; (**d**) comparison of pediatric emergency visits per area between periods.

**Figure 6 children-09-01464-f006:**
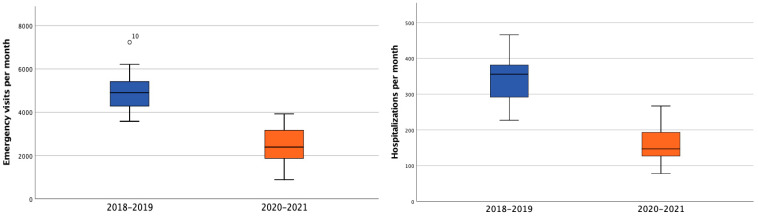
Comparison of pediatric emergency visits and hospitalizations per month during the periods 2018–2019 (blue) and 2020–2021 (orange).

**Table 1 children-09-01464-t001:** Comparison of the detection rate of eight respiratory viruses between the periods 2018–2019 and 2020–2021.

	Detection Rate by Year (%); 95% CI		
	March 2018–September 2019	March 2020–September 2021	*p*-Value
**Total**	54.1 (53.1–55.0)	20.1 (19.4–20.7)	<0.001
**Non-enveloped viruses**			
Adenovirus	21.3 (20.5–22.1)	8.6 (8.2–9.1)	<0.001
Rhinovirus	2.9 (2.6–3.2)	0.5 (0.4–0.6)	<0.001
Enterovirus	11.1 (10.5–11.7)	2.5 (2.3–2.8)	<0.001
Paraechovirus	0.1 (0.03–0.14)	0.4 (0.3–0.5)	<0.001
**Enveloped viruses**			
Influenza	10.3 (9.8–10.9)	0.3 (0.2–0.4)	<0.001
Respiratory syncytial virus	7.9 (7.4–8.5)	2.4 (2.1–2.7)	<0.001
Human coronavirus	3.6 (3.3–4.0)	1.7 (1.5–1.9)	<0.001
Metapneumovirus	1.7 (1.5–2.0)	0.3 (0.2–0.4)	<0.001
Parainfluenza	1.7 (1.5–2.0)	0.2 (0.1–0.3)	<0.001

## Data Availability

The datasets generated during and analyzed during the current study are available from the corresponding author on reasonable request.

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
