# Peer review of "Change on the Circulation of Respiratory Viruses and Pediatric Healthcare Utilization during the COVID-19 Pandemic in Asturias, Northern Spain"

_children, 2022, doi:10.3390/children9101464_

Round 1

Reviewer 1 Report

Dear author, 

Your article Change on the circulation of respiratory viruses and pediatric 2 healthcare utilization during the COVID-19 pandemic in Asturias, northern Spain is very interesting 

My decision accept. 

Author Response

Dear reviewer, 

thank you for your comments on my paper

Reviewer 2 Report

Comments to the Author:

The article entitled “Change on the circulation of respiratory viruses and pediatric healthcare utilisation during the COVID-19 pandemic in Asturias, northern Spain” provides a discussion on the respiratory viral circulation pattern in children in Asturias during the early COVID-1 pandemic and the comparison of the number of pediatric hospitalisation and emergency visits between the same period before and during the pandemic. This article gives another valuable insight into the impact of the SARS-CoV-2 pandemic and its control measures on the circulation of other respiratory viruses and pediatric hospitalisation. Furthermore, the research questions are well defined, followed by comprehensive analysis and discussion. This article is generally well-written and structured.

1.     The title of the review is clear.

2.      The Introduction is relevant to the other sections of this article. In the results and discussion, the authors use the terms “enveloped and non enveloped viruses”, so it would be better if they added a few related sentences. Moreover,  the authors may add a few sentences related to public health policy that can be proposed based on the results of this study.

3.     Methodology:

The methodology section is clear, separating SARS CoV-2 and other respiratory viruses.

3.1  In line 102, the authors should delete “COVID-19 Pandemic.”

4.     Results

4.1  The authors may only show 1 figure with section 3.1 instead of 2 figures. Figure 3 may be provided as a supplementary figure.

4.2. As the authors did not discuss details regarding the difference in children hospitalisation between health areas, it would be better if figures 5c and 5d were provided as supplementary figures and replaced with figure 6.

5.     Discussion

The discussion is comprehensive and well-written. The authors may add recommendations related to public health policy and the future study needed, especially in the conclusion.

Author Response

Dear Reviewer,

Thank you for your comments on my paper. Here I attach my responses and comments on the changes made on the manuscript:

  1. The title of the review is clear.

  1. The Introduction is relevant to the other sections of this article. In the results and discussion, the authors use the terms “enveloped and non enveloped viruses”, so it would be better if they added a few related sentences. Moreover,the authors may add a few sentences related to public health policy that can be proposed based on the results of this study.

A few sentences have been added on this matter (lines 64-67)

  1. Methodology:

The methodology section is clear, separating SARS CoV-2 and other respiratory viruses.

3.1  In line 102, the authors should delete “COVID-19 Pandemic.”

This has been deleted

  1. Results

4.1  The authors may only show 1 figure with section 3.1 instead of 2 figures. Figure 3 may be provided as a supplementary figure.

4.2. As the authors did not discuss details regarding the difference in children hospitalisation between health areas, it would be better if figures 5c and 5d were provided as supplementary figures and replaced with figure 6.

Even though we appreciate the comments on the results, we would like to keep the figures on the text and not on the supplementary material, because they explain different information and help a deeper understanding of the research content for readers.

  1. Discussion

The discussion is comprehensive and well-written. The authors may add recommendations related to public health policy and the future study needed, especially in the conclusion.

A few sentences have been added in the conclusion (lines 305-306), even though this is also explained in the discussion (278-282)

Reviewer 3 Report

This paper shows a large reduction in non-COVID-19 respiratory viruses and hospital use in children in 2018-2019 (pre-pandemic) and 2020-2021 (pandemic) in the Asturias region, Spain. This has been previously well described including in a larger national Spanish study. This is a straightforward descriptive study of the reduced numbers which does not attempt to analyse or discuss the underlying reasons in depth. There are few minor grammatical errors which need correcting.

Lines 99-100: "Influenza virus, RSV, Adenovirus, Enterovirus, Parainfluenzavirus, Metapneumovirus, Rhinovirus, Parechovirus, Human coronavirus" - no need for capital letters for virus names. Please change throughout paper.

Line 95: "inhouse multiplex RT-PCR- real time" should be "in-house multiplex real-time RT-PCR". Give a citation for this assay if available; if not available, please give some indication that this in-house assay has been validated or has been approved for use by a local regulatory agency. This is important to give confidence to the reader that this key part of the study results is reliable.

Line 100: "Human coronavirus" - which ones?

Fig 1: vertical axis should specify incidence per 100,000

Fig 2: Instead of showing "number of viral detections", which is affected by number of samples sent, it may be better to show "% of virus detection" (e.g. RSV-positive samples/total samples sent).

Fig 3: mention in the figure label what the enveloped and non-enveloped viruses are.

Fig 5: horizontal axes - names of months should be capitalized

For Figs 4-6, I suggest use the same colour consistently, e.g. orange for 2020-2021, blue for 20118-2019, to make it easier for readers.

Line 234: "de" should be "the"

Lines 262-263: Please expand why you think stability and resistance to disinfectants explains the difference in detection of enveloped/non-enveloped viruses.

Lines 265-266: can you suggest reasons why influenza and RSV was undetected in the 2021 winter when schools/kindergartens were already open, as schools are known to be an important source of community transmission of influenza? Were schools fully open and attended in Asturias, or were there a lot of optional on-line classes and reduced attendance?

The unusual RSV season has been described in other countries and warrants more discussion than has been presented here.

Author Response

Dear Reviewer,

Thank you for your comments on my paper. Here I attach my responses and comments on the changes made on the manuscript:

Lines 99-100: "Influenza virus, RSV, Adenovirus, Enterovirus, Parainfluenzavirus, Metapneumovirus, Rhinovirus, Parechovirus, Human coronavirus" - no need for capital letters for virus names. Please change throughout paper.

This has been changed throughout the paper

Line 95: "inhouse multiplex RT-PCR- real time" should be "in-house multiplex real-time RT-PCR". Give a citation for this assay if available; if not available, please give some indication that this in-house assay has been validated or has been approved for use by a local regulatory agency. This is important to give confidence to the reader that this key part of the study results is reliable.

This has been changed (lines 102-103)

Line 100: "Human coronavirus" - which ones?

I have specified that on the text (NL63, OC43, 229E, HKU21) (line 109)

Fig 1: vertical axis should specify incidence per 100,000

This has been added

Fig 2: Instead of showing "number of viral detections", which is affected by number of samples sent, it may be better to show "% of virus detection" (e.g. RSV-positive samples/total samples sent).

Even though we appreciate this comment, the authors have decided on this figure because the % of virus detection is already shown on table 1, and the figure is more clear showing number of viral detections.

Fig 3: mention in the figure label what the enveloped and non-enveloped viruses are.

This has been added in the figure label

Fig 5: horizontal axes - names of months should be capitalized

This has been corrected

For Figs 4-6, I suggest use the same color consistently, e.g. orange for 2020-2021, blue for 2018-2019, to make it easier for readers.

The colors have been changed

Line 234: "de" should be "the"

Line 260: corrected

Lines 262-263: Please expand why you think stability and resistance to disinfectants explains the difference in detection of enveloped/non-enveloped viruses.

A few sentences have been added, according to the previous reports on this topic (lines 300-301)

Lines 265-266: can you suggest reasons why influenza and RSV was undetected in the 2021 winter when schools/kindergartens were already open, as schools are known to be an important source of community transmission of influenza? Were schools fully open and attended in Asturias, or were there a lot of optional on-line classes and reduced attendance?

Schools were fully open and attended in Asturias in the 2021 winter. There were no online classes anymore or reduced attendance at the time. The reasons why influenza and RSV were undetected in the 2021. As explained in the introduction and getting to the same conclusion as other studies, it could be related to the non-pharmaceutical interventions as face mask, contact tracing or reduced of large groups, or it may be related to viral interference. The results of our descriptive study can no go further on these conclusions.

The unusual RSV season has been described in other countries and warrants more discussion than has been presented here.